# Omeprazole Inhibits Glioblastoma Cell Invasion and Tumor Growth

**DOI:** 10.3390/cancers12082097

**Published:** 2020-07-28

**Authors:** Un-Ho Jin, Sharon K. Michelhaugh, Lisa A. Polin, Rupesh Shrestha, Sandeep Mittal, Stephen Safe

**Affiliations:** 1Department of Veterinary Physiology and Pharmacology, College of Veterinary Medicine, Texas A&M University, College Station, TX 77843, USA; jinunho@gmail.com; 2Fralin Biomedical Research Institute, Virginia Tech Carilion School of Medicine, Roanoke, VA 24014, USA; skmichel@vtc.vt.edu (S.K.M.); sandeepmittal@vt.edu (S.M.); 3Department of Oncology, Wayne State University and Karmanos Cancer Institute, Detroit, MI 48201, USA; polinl@karmanos.org; 4Department of Biochemistry and Biophysics, Texas A&M University, College Station, TX 77843, USA; rshrestha@tamu.edu; 5Carilion Clinic-Neurosurgery, Roanoke, VA 24014, USA

**Keywords:** patient-derived xenograft, glioblastoma, selective AhR modulators, growth inhibition, cell invasion

## Abstract

*Background*: The aryl hydrocarbon receptor (AhR) is expressed in gliomas and the highest staining is observed in glioblastomas. A recent study showed that the AhR exhibited tumor suppressor-like activity in established and patient-derived glioblastoma cells and genomic analysis showed that this was due, in part, to suppression of CXCL12, CXCR4 and MMP9. *Methods*: Selective AhR modulators (SAhRMs) including AhR-active pharmaceuticals were screened for their inhibition of invasion using a spheroid invasion assay in patient-derived AhR-expressing 15-037 glioblastoma cells and in AhR-silenced 15-037 cells. Invasion, migration and cell proliferation were determined using spheroid invasion, Boyden chambers and scratch assay, and XTT metabolic assays for cell growth. Changes in gene and gene product expression were determined by real-time PCR and Western blot assays, respectively. In vivo antitumorigenic activity of omeprazole was determined in SCID mice bearing subcutaneous patient-derived 15-037 cells. *Results*: Results of a screening assay using patient-derived 15-037 cells (wild-type and AhR knockout) identified the AhR-active proton pump inhibitor omeprazole as an inhibitor of glioblastoma cell invasion and migration only AhR-expressing cells but not in cells where the AhR was downregulated. Omeprazole also enhanced AhR-dependent repression of the pro-invasion CXCL12, CXCR4 and MMP9 genes, and interactions and effectiveness of omeprazole plus temozolomide were response-dependent. Omeprazole (100 mg/kg/injection) inhibited and delayed tumors in SCID mice bearing patient-derived 15-037 cells injected subcutaneously. *Conclusion*: Our results demonstrate that omeprazole enhances AhR-dependent inhibition of glioblastoma invasion and highlights a potential new avenue for development of a novel therapeutic mechanism-based approach for treating glioblastoma.

## 1. Introduction

An estimated 23,880 new cases of cancer of the brain and nervous system are diagnosed each year and 16,380 deaths will occur from these diseases annually [1,2]. Glioblastoma (GBM) is the most frequently diagnosed malignant primary brain tumor and global incidence of this disease varies from 0.59–3.69 per 100,000 [3]. A diagnosis of GBM in an adult is devastating since patient survival times are in the range of 12-15 months and the 3-year survival of patients after diagnosis remains a dismal 3–5% [4,5,6,7]. Primary de novo GBMs constitute approximately 90% of all cases and occur in elderly patients, whereas secondary GBMs that progress from lower grade gliomas are mainly diagnosed in younger patients [8]. Glioblastoma is a complex disease which involves multiple genetic alterations including mutations of multiple genes such as *p53*, epidermal growth factor receptor (*EGFR*), platelet-derived growth factor receptor (*PDGFR*), isocitrate dehydrogenase-1 (*IDH1*), mouse double minute homolog 2 (*MDM2*), and phosphate and tensin homolog (*PTEN*) [9,10,11,12,13,14,15]. Many of the genetic alterations are associated with loss of heterozygosity of the chromosome arm 10q, which is observed in 60-90% of GBM cases [16,17]. These genetic changes result in a complex and highly aggressive disease which is difficult to treat and the current standard-of-care for newly-diagnosed GBM that includes surgery, adjuvant radiotherapy and the drug temozolomide (TMZ; an alkylating agent) have limited success [8,18]. Thus, there is a dire need to further understand the molecular underpinnings of this disease and develop new mechanism-based agents that are effective against GBM and can be used in individual and combined therapies.

The AhR is a ligand-activated receptor that was initially identified as the receptor that bound the highly toxic 2,3,7,8-tetrachlorodibenzo-*p*-dioxin (TCDD) and structurally related halogenated aromatics and mediated the downstream toxic effects induced by these compounds [19,20]. The AhR nuclear translocator (Arnt) protein forms a transcriptionally active nuclear heterodimer with the AhR and binds *cis*-acting xenobiotic response elements (XREs) in target gene promoters to activate Ah-responsive genes such as CYP1A1 [21]. Subsequent studies showed that the AhR acts through multiple pathways including both genomic and non-genomic (largely Arnt-independent) and binds structurally diverse compounds including endogenous molecules such as tryptophan metabolites, pharmaceuticals, and health-promoting phytochemicals [22,23]. Moreover, there is evidence that the AhR and its ligands play an important role in maintaining cellular homeostasis and in pathophysiology [24,25,26,27].

Immunostaining of the AhR in infiltrating gliomas from 73 patients showed that the AhR was expressed in all tumors with the highest staining observed in GBM [28]. Established GBM cell lines also express the AhR, however, our recent study [29] identified a patient-derived glioma cell line (PDG 14-015s) which did not express the AhR. Previous reports demonstrated that the AhR exhibited pro-oncogenic activity in medulloblastomas and pituitary adenomas and also in GBM cells [30,31,32,33,34]. Moreover, Opitz and co-workers reported that the tryptophan metabolite kynurenine enhanced GBM cell growth and blocked immune surveillance of the tumor. In contrast, the AhR enhances differentiation in some neuroblastoma cells and treatment of PC12 cells with TCDD induces apoptosis [35,36]. Our recent study in established and patient-derived GBM cell lines [37] showed that neither TCDD nor kynurenine affected growth or invasion of GBM cells. Moreover, genomic and functional studies demonstrated that the AhR exhibited tumor suppressor-like activity and inhibited GBM invasion [29]. Previous studies show that omeprazole activates AhR-mediated responses, induces nuclear uptake of the AhR and transforms cytosolic AhR to bind oligonucleotides containing an XRE but there is no evidence that omeprazole directly interacts with the receptor ligand binding domain [38,39,40,41,42,43]. It is possible that omeprazole activates the receptor to bind an as yet unidentified endogenous ligand [43] or binds other sites on the AhR. Studies in this laboratory have previously reported that omeprazole inhibits invasion of breast and pancreatic cancer cells and these responses were AhR-dependent [44,45]. In this study, we show that the AhR-active proton pump inhibitor omeprazole (OME) acts as a selective AhR modulator (SAhRM) and decreases growth and invasion of GBM cells in culture and inhibits tumor growth in vivo.

## 2. Materials and Methods

### 2.1. Cell Lines, Antibodies, and Reagents

U87-MG human malignant glioma cell line was obtained from the American Type Culture Collection (Manassas, VA, USA). Patient-derived cell lines 14-015s (GBM), 14-104s (grade IV gliosarcoma), and 15-037 (GBM) were generated from fresh tumor specimens collected from newly diagnosed patients with no prior chemo- or radiotherapy treatment as previously described [29]. All glioma cells were maintained in Dulbecco’s Modified Eagle’s Medium (DMEM)/Hams F-12 50/50 mix supplemented with L-glutamine, 10% fetal bovine serum (FBS), 1X MEM non-essential amino acids, and 10 µg/ml gentamycin (Gibco) as described in [29]. Antibodies and reagents are summarized in Appendix A. The doses of OME, esomeprazole (ESO), and other AhR ligands in the invasion/migration/growth assays were the highest concentrations that were not cytotoxic (i.e., <15% cell death) and transfection experiments were carried out as described [29].

### 2.2. Quantitative Real-Time PCR

RNA was isolated using the Quick-RNA MiniPrep Kit (Zymo Research, Irvine, CA, USA). Quantification of mRNA was performed using iTaq Universal SYBR Green 1-Step Kit (Bio-Rad Laboratories, Hercules, CA, USA) using the manufacturer’s protocol with the CFX384 real-time PCR System (Bio-Rad Laboratories, Hercules, CA, USA). The comparative CT method was used for relative quantitation of samples. Values for each gene were normalized to expression levels of TATA-binding protein (TBP) as described in [29]. The sequences of the primers used for real-time PCR are summarized in Appendix A.

### 2.3. Scratch, Invasion and Proliferation Assays

After cells were more than 80% confluent in 6-well plates, a scratch was made using a sterile pipette and cell migration into the scratch was determined after 18 hours. The BD-Matrigel Invasion Chamber (24-transwell with 8 μm pore size polycarbonate membrane) was used in a modified Boyden chamber and the 3-D tumor spheroid invasion assay and XTT cell proliferation assays were carried out as previously reported [29].

### 2.4. In Vivo Tumor Growth Assay

This study was approved by the Wayne State University Institutional Animal Care and Use Committee (#17-08-0315). Development of the 15-037 GBM patient-derived xenograft (PDX) model has been previously described [37]. Eight-week old female SCID mice (C.B-17/IcrHsd-Prkdcscid; Envigo RMS, Inc., Indianapolis, IN) were supplied food and water ad libitum. Tumor fragments from GBM 15-037 (~30 mg; passage 17) were implanted subcutaneously and bilaterally via trocar on study day 0. Mice were randomized into 2 groups: OME or vehicle control (*n* = 6 each). Omeprazole (catalog # O104, MilliporeSigma, St. Louis, MO, USA) was suspended in olive oil and administered daily at 100 mg/kg per injection (0.1 mL) by oral gavage beginning on study day 3. On study day 21, dosing was escalated to twice daily, with the same 100 mg/kg dose per injection with the last injection on study day 27 for a total of 31 injections. Estimated tumor volume was assessed 2–3× weekly by caliper measurements and calculated with the formula (length × width^2^)/2. Body weights were measured daily. Mice were euthanized when they reached the study endpoint (estimated tumor volume ~5% of total body weight). Endpoints for antitumor efficacy analysis have been previously published [46] and are summarized here:

### 2.5. %T/C Value (Inverse of Tumor Growth Inhibition

%*T*/*C* = *T*/*C* × 100: (T) is the median tumor weight in the treated group and (C) is the median tumor weight of the control group at a selected time point when control tumors are in exponential growth phase (range: 750–1250mm^3^). Results are expressed as percentage.

### 2.6. Tumor Growth Delay (T-C Value)

(T) is the median value in days required for the treatment group tumors to reach a predetermined size (e.g., 750 mm^3^), and (C) is the median time in days for the control group tumors to reach the same size. This metric is crucial because it allows the quantification of tumor-cell kill.

### 2.7. Calculation of Tumor Cell Kill (or Log Cell Kill)

For subcutaneously growing tumors, the log10 cell-kill is calculated from the following formula: Gross log cell kill (GLK, LCK) = T−C value in days/3.32 (Td) where (T−C) is the tumor growth delay described above and Td is the tumor volume doubling time (in days), estimated from the best fit straight line from a semi-log plot of the control group tumors in exponential growth phase.

### 2.8. Tumor Tissue Staining

Formalin-fixed tumor sections were stained with hematoxylin and eosin (H&E) or immunostained with an antibody that recognized AhR (cat# ab84833, Abcam, Cambridge, MA, USA) following previously published protocols [28,37]. All animal studies were carried out according to the procedures approved by the Wayne State University and Virginia Tech Carillon School of Medicine Institutional Animal Care and Use Committees.

### 2.9. Statistics

All experiments were repeated a minimum of three times. The data are expressed as the means (SD). Statistical significance was analyzed using either unpaired Student’s t-test (two-tailed) or analysis of variance (ANOVA) test. For the in vivo study, descriptive statistics and unpaired Student’s t-test (one-tailed) were calculated with GraphPad Prism v. 8.4.2 for Windows, GraphPad Software (www.graphpad.com, San Diego, CA, USA). For all experiments, a p value of less than 0.05 was considered statistically significant.

## 3. Results

The AhR is expressed in both established GBM cell lines and PDG cells and results of knockdown studies show that the AhR inhibits migration/invasion in GBM cells and we also observed that neither kynurenine or TCDD affected GBM cell invasion [29]. In this study, we screened a series of AhR-active compounds including pharmaceuticals to determine their effects on a spheroid invasion in patient-derived 15-037 cells (wild-type) and cells in which the AhR has been knocked down by RNA interference (RNAi) (15-037 siAhR) or by CRISPR/Cas9 (15-037 AKO-19). In solvent-treated cells, knockdown of AhR in 15-037 by RNAi (siAhR) and in 15-037 AKO-19 cells resulted in a significant increase in invasion compared to cells expressing AhR (Figure 1A). We next investigated the effects of several AhR-active pharmaceuticals including leflunomide (Figure 1B), mexiletine (Figure 1C), tranilast (Figure 1D), β-naphthoflavone (Figure 1E), the AhR active microbial metabolite tryptamine (Figure 1F), and OME, a pharmaceutical that inhibits invasion of breast and pancreatic cancer cells [44,45] (Figure 1G). In this study, most of the compounds inhibited invasion in control (AhR^+/+^) cells with the exception of leflunomide in 15-037 wild-type cells (Figure 2B, right panels). However, only OME and trypamine (Figure 1F,G) did not affect invasion in both 15-037 knockout cell lines. Interestingly, the remaining compounds inhibited invasion in AhR-expressing and AhR knockout cells, indicating that these responses were AhR-independent.

Therefore, we used OME as model AhR-active compound and further investigated effects on GBM cell invasion and the results in Figure 2A show that both OME and ESO (an isomer of OME) decreased invasion in 15-037 cells. Loss of AhR (by RNAi) increased invasion and attenuated the inhibitory effects of both compounds. In a previous study, we showed that multiple siRNAs decrease AhR in 15-037 cells and one of the oligonucleotides [29] was used in this study and efficiently decreased AhR levels (Figure 2B) and also decreased CYP1A1 expression in 15-037 cells treated with DMSO, OME, and ESO (Figure 2C). CYP1A1 has been used as a biochemical marker of Ah-responsiveness and we observed that TCDD significantly induced CYP1A1 mRNA levels in U87, and PDG cells 14-104s and 15-037 but not in AhR-deficient 14-015s cells [29] (Figure 2D). In contrast, OME and ESO were inactive as CYP1A1 inducers in PDG cell lines but induced a 2–3-fold response in U87 cells. Appendix A compares the effects of TCDD vs. OME and ESO as inducers of 3 Ah-responsive genes (CYP1A1, CYP1B1, and TiPARP) in 3 Ah-responsive GBM cell lines (U87, 14-104s, and 15-037). The results show that TCDD was active as an inducer for all responses, whereas the effects of OME and ESO were response and cell context dependent.

Previous genomic and mechanistic studies showed that CXCL12, CXCR4, and MMP9 were critical pro-invasion genes repressed by the AhR [29] in GBM cells and both OME and ESO decreased expression of all three genes in wild-type 15-037 cells (Figure 2E). Knockdown of the AhR in these cells attenuated the inhibitory effects of OME and ESO on all three genes. However, in the absence of the AhR, OME strongly induced CXCR4 mRNA levels and the reason for this unexpected response is currently being investigated. We also investigated the effects of these treatments on NFkB (p65) phosphorylation and the loss of AhR decreased p65 but omeprazole did not affect p65 levels in cells (±AhR) (data not shown). We also compared the effects of OME and ESO in 15-037 (wild-type) and 15-037 AKO-19 cells and the results (Figure 3) complemented studies illustrated in Figure 2. OME and ESO decreased invasion in the spheroid invasion assay in 15-037 cells but not on AhR-deficient cells (Figure 3A). Figure 3B illustrates the efficiency of AhR knockdown by CRISPR/Cas9 [29]. OME and ESO did not induce CYP1A1 (Figure 3C) and both OME and ESO decreased expression of the pro-invasion genes CXCL12, CXCR4, and MMP in wild-type but not in cells depleted of the AhR. We also investigated the effects of OME and ESO on AhR-dependent invasion of U87 and 14-104s cells (Figure 4A), as well as 14-015s and 15-037 cells (Figure 4B) using the Boyden chamber assay. Quantitation of the results (Figure 4C) showed that 100 µM OME decreased invasion in U87, 14-104s, and 15-037 cells; and, with the exception of 100 µM ESO in U87 cells, the inhibition of invasion by OME and ESO was attenuated after AhR knockdown. It should also be noted that the AhR-depleted 14-015s cells are more invasive than the AhR expressing 14-104s and 15-037 patient-derived cells (Figure 4B) and this is consistent with the inhibitory effects of the AhR on cell invasion [29]. Figure 4D shows that siAHRs efficiently decreased AhR levels in both established and patient-derived GBM cells. We also showed that TCDD did not affect invasion and that both OME and ESO inhibited invasion (AhR-dependent) in another established GBM cell line (T98G cells) (Appendix A). ESO was less potent than OME in the Boyden chamber assay. The effects of OME and ESO in 14-015s were AhR-independent, which is consistent with previous studies showing that this cell line expresses minimal levels of the AhR [29].

TMZ is part of the standard-of-care for treatment of GBM and we, therefore, investigated possible interactions between TMZ and OME in invasion, migration, and cell proliferation assays in 15-037 cells. In a scratch assay, we observed that 500 µM TMZ and 100 µM OME inhibited cell migration and the combination of both drugs enhanced the inhibitory response (Figure 5A). TMZ and OME inhibited invasion in a Boyden chamber assay (Figure 5B); however, the combination of both drugs did not enhance the inhibitory response. This was in contrast to the spheroid invasion assay results (Figure 5C) where OME inhibited invasion and TMZ alone or in combination with OME were inactive as inhibitors. Using this same approach, we observed that both TMZ and OME inhibited 15-037 cell proliferation and the drug combination further enhanced the inhibitory response after 3 days. These results confirm that OME inhibits 15-037 cell growth, migration, and invasion. Moreover, when combined with TMZ, the effectiveness of OME is response-dependent.

We also investigated the in vivo effects of OME on tumor growth in SCID mice bearing subcutaneous 15-037 PDX tumors. One mouse in the no treatment control group failed to develop tumors and was excluded from study (it failed subsequent re-challenge with the same tumor and was determined to be a spontaneous “leaky” SCID). Mice in the OME-treated group tolerated the 100 mg/kg/injection dosing with no overt toxicity. Body weights were consistent between groups throughout the study (Appendix A). OME inhibited and delayed tumor growth (Appendix A; Figure 6). At the time of the final administration of OME, study day 27, tumors were significantly smaller in the treated group (Figure 6A). Additionally, when control tumors reached the exponential growth phase (~750–1250 mg, study day 35), the % T/C = 42% or tumor growth inhibition (TGI) was 58% (Appendix A), which was also demonstrated as a tumor growth delay (T-C) of 8 days compared to the control tumor group, i.e., the time for tumors to reach 750 mg in the OME-treated mice (Figure 6B, Appendix A). OME treatment produced an overall 0.7 logs of gross log10 cell kill, but once treatment stopped, tumor burden in the OME group progressed and reached the study endpoint of 5% body mass and mice were euthanized. Tumor tissue H&E stains (Figure 6C,D) revealed a similar tumor cell density in both the control and OME-treated mice. AhR immunostaining (Figure 6E,F) demonstrated that AhR localization was cytoplasmic, and not nuclear, which would be expected as mice were sacrificed a minimum of 7 days after the last dose of OME was administered.

## 4. Discussion

The role of the AhR and its ligands have been extensively investigated and there is evidence that AhR expression exhibits both tumor-promoting and tumor-suppressive activities and this is dependent on the tumor type. However, like other receptors, relatively non-toxic SAhRMs which exhibit tissue-specific AhR agonist or antagonist activities have been developed [47,48] including the AhR-active drug “aminoflavone” which has been in clinical trials for breast cancer [49]. Many commonly used pharmaceuticals exhibit AhR activity [50] and several of these compounds including tranilast, raloxifene, leflunomide, OME, and flutamide have been characterized as SAhRMs and inhibited tumor growth [44,45,51,52,53]. Studies in this laboratory have also focused on repositioning AhR pharmaceuticals for chemotherapeutic applications in both breast and pancreatic cancer [44,45] and have demonstrated that inhibition of tumor invasion and metastasis by OME and tranilast was AhR-dependent by tumor-specific mechanisms. In triple-negative breast cancer cells, OME inhibits invasion in vitro and lung metastasis in vivo through a genomic pathway involving downregulation of the G-protein coupled receptor CXCR4, which is involved in tumor promotion and metastasis [44]. In this study, we initially screened several AhR ligand as inhibitors of spheroid cell invasion using Ah-responsive patient-derived 15-037 cells in which the AhR is transiently decreased by RNAi or permanently knockdown by CRISPR/Cas9 [29]. Many of the compounds decreased invasion in wild-type and AhR-silenced 15-037 cells (Figure 1); however, only tryptamine (an AhR-active microbial metabolite of tryptophan) and OME induced AhR-dependent spheroid cell invasion.

Previous studies demonstrated that OME inhibited invasion of breast and pancreatic cancer cells [53]. We, therefore, used OME and its isomer, ESO, as models for investigating their ability to inhibit GBM migration and invasion. Results illustrated in Figure 2 and Figure 3, using Ah-responsive 15-037 cells in which the AhR is knocked down either transcently by RNAi or permanently by CRISPR/Cas9 [29], showed complementary effects induced by OME and ESO. Both compounds decrease AhR-dependent invasion in the spheroid invasion assay and related pro-invasion genes (CXCL12, CXCR4, and MMP9). NFkB plays an important role in the invasion of glioblastoma cells [54] and there is also evidence that the AhR suppresses NFkB [55,56]. We examined effects of AhR silencing and treatment with omeprazole on NFkB-p65 and showed that loss of AhR decreased p65 and OME did not affect p65 levels (±AhR) (data not shown). Induction of CYP1A1, a widely used biomarker for AhR-active compounds and other Ah-responsive genes, was minimal to non-detectable in a panel of GBM cell lines treated with OME and ESO (Figure 2D). The observation that TCDD induced CYP1A1 but did not inhibit GBM invasion and that OME inhibited invasion but did not induce or minimally induced other AhR-responsive genes (CYP1A1, CYP1B1, and TiPARP) (Appendix A) suggests that OME is a SAhRM that selectively modulates AhR-mediated responses. Similar differences have been reported for TCDD and OME in breast and pancreatic cancer cells [44,45].

We also observed that OME and ESO inhibited GBM invasion not only in the tumor spheroid assay but also in the Boyden chamber assay and this was complemented by inhibition of cell migration (Figure 5A) and proliferation (Figure 5D). We also compared the effects of combined drugs (500 µM TMZ and 100 µM OME) in the tumor spheroid, Boyden chamber scratch, and cell proliferation assays. With the exception of the tumor spheroid assay for TMZ, both compounds were inhibitors; however, their combined effects were enhanced only in the scratch and cell proliferation assays. We are currently extending our studies on OME/TMZ interactions over a range of concentrations and cell lines, and using a genomic approach to determine pathway/gene specifity for interactions of these compounds.

We also investigated the in vivo effects of OME. In the GBM PDX mouse model, OME administration modestly inhibited and delayed tumor growth. Based on our previous study [29] and in vitro data presented here, activation of AhR by OME suppressed proliferation of the GBM cells, reducing tumor mass and slowing tumor growth. After discontinuation of OME, the PDX tumors resumed a growth rate in parallel to the tumors from the control mice. In this study, OME did have the advantage of being well-tolerated by the mice with minimal overt toxicity observed. However, OME is relatively insoluble and may be inconsistently absorbed via the oral gavage route that was employed leading to the high dose requirement for the observed activity. These limitations prevented OME from generating a sustained response in the suppression of tumor growth. Nevertheless, OME demonstrated sufficient inhibition of tumor growth in the GBM PDX model based on %T/C, T-C, and log cell kill analysis. These findings suggest that SAhRMs with increased solubility and potency or when used in combination with TMZ would likely have greater efficacy and present a new opportunity for the development of novel therapeutic avenues for GBM patients.

## 5. Conclusions

In summary, our results show that omeprazole inhibits glioblastoma cell invasion and this is similar to results of previous studies in breast and pancreatic cancer cells [44,45]. The effects of omeprazole on glioblastoma cell invasion are AhR-dependent but that does not preclude contributions of other pathways Several review articles summarize the anticancer activities and safety concerns of omeprazole and related proton pump inhibitors and their effects are variable [57,58,59,60,61]. However, our results identify the AhR as a viable drug target for treating glioblastoma and this could include specific SAhRMs and also other AhR-active benzimidazole [62] that might mimic the effects of omeprazole.

## Figures and Tables

**Figure 1 cancers-12-02097-f001:**
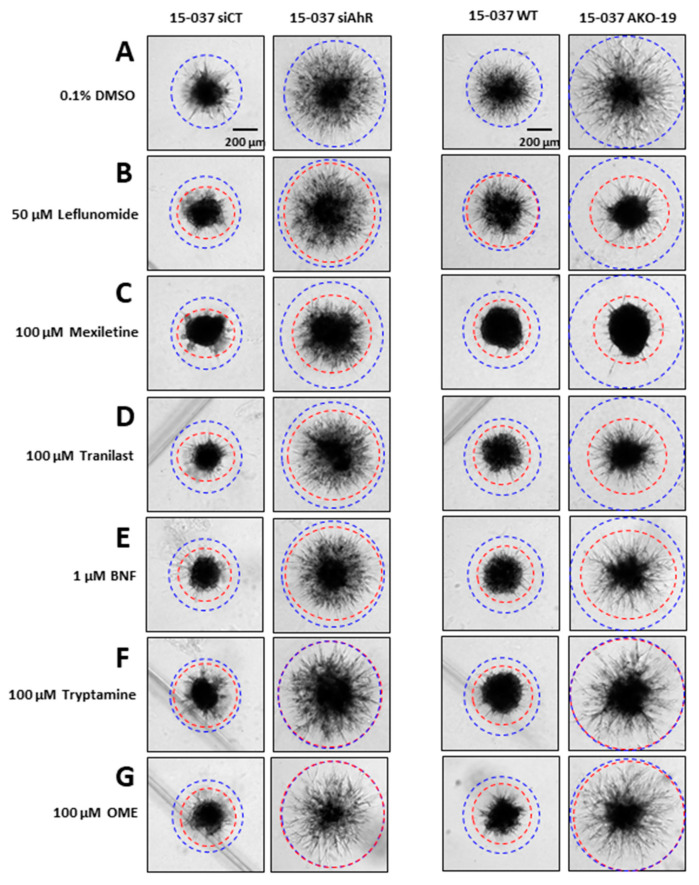
AhR-active compound screening. Wild-type 15-037 patient-derived Glioblastoma (GBM) cells (AhR^+/+^) or cells knockdown for the AhR by RNAi or CRISPR/Cas9 (AKO-19) were treated with DMSO (**A**), leflunomide (**B**), mexiletine (**C**), tranilast (**D**), β-naphthoflavone (BNF) (**E**), tryptamine (**F**), and omeprazole (OME) (**G**) in a spheroid invasion assay. Compound-induced invasion in wild-type and AhR knockdown cells were compared to solvent (DMSO) treated cells and indicated directly in each panel.

**Figure 2 cancers-12-02097-f002:**
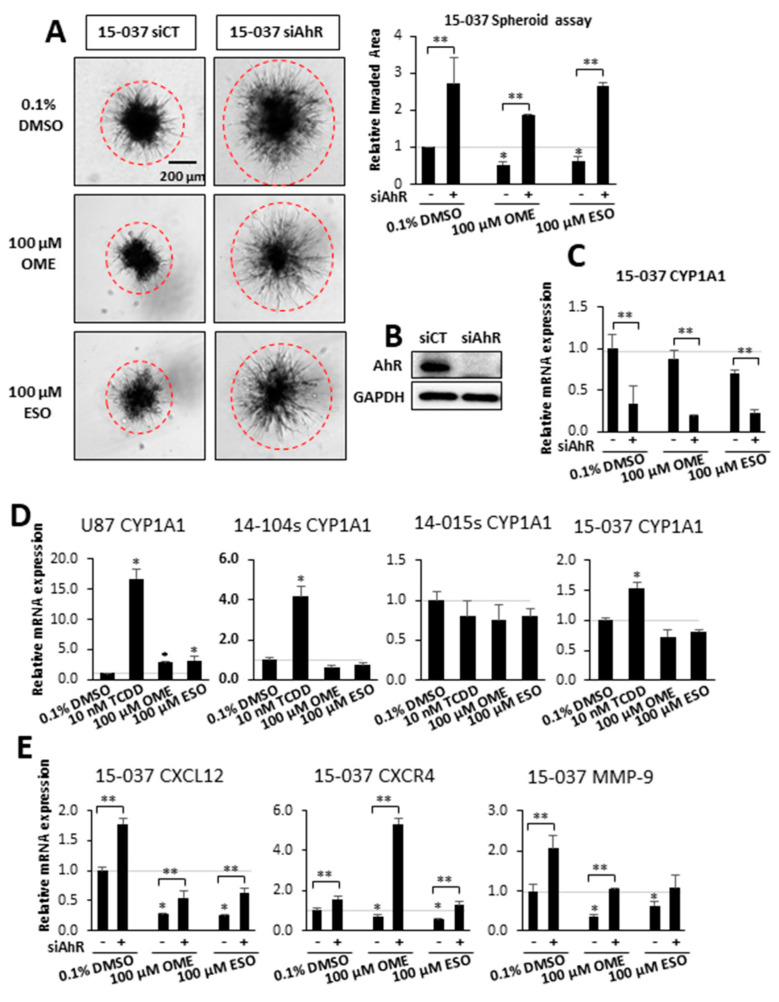
Effects of OME and esomeprazole (ESO) on AhR-dependent invasion and gene expression in wild-type 15-037 siAhR cells. Wild-type 15-037 and AhR-deficient 15-037 cells were treated with DMSO, 100 µM OME, and 100 µM ESO and spheroid cell invasion was determined and quantitated (**A**). AhR expression after knockdown by siAhR in 15-037 cells was determined by Western blot of whole cell lysates (**B**). Effects of OME and ESO on CYP1A1 expression in wild-type and AhR-deficient 15-037 cells (**C**), effects of 2,3,7,8-tetrachlorodibenzo-*p*-dioxin (TCDD), OME, and ESO in CYP1A1 induction in established and patient-derived GBM cells (**D**), and effects of OME and ESO on CXCL12, CXCR4, and MMP9 expression in wild-type and AhR-deficient 15-037 cells (**E**) was determined by real time PCR as outlined in the Materials and Methods. Results (**A**,**C**–**E**) are means ± SD for at least 3 determinations for each treatment group and significant (*p* < 0.05) effects by the compounds alone (*) and attenuation of these responses after AhR knockdown (**) are indicated.

**Figure 3 cancers-12-02097-f003:**
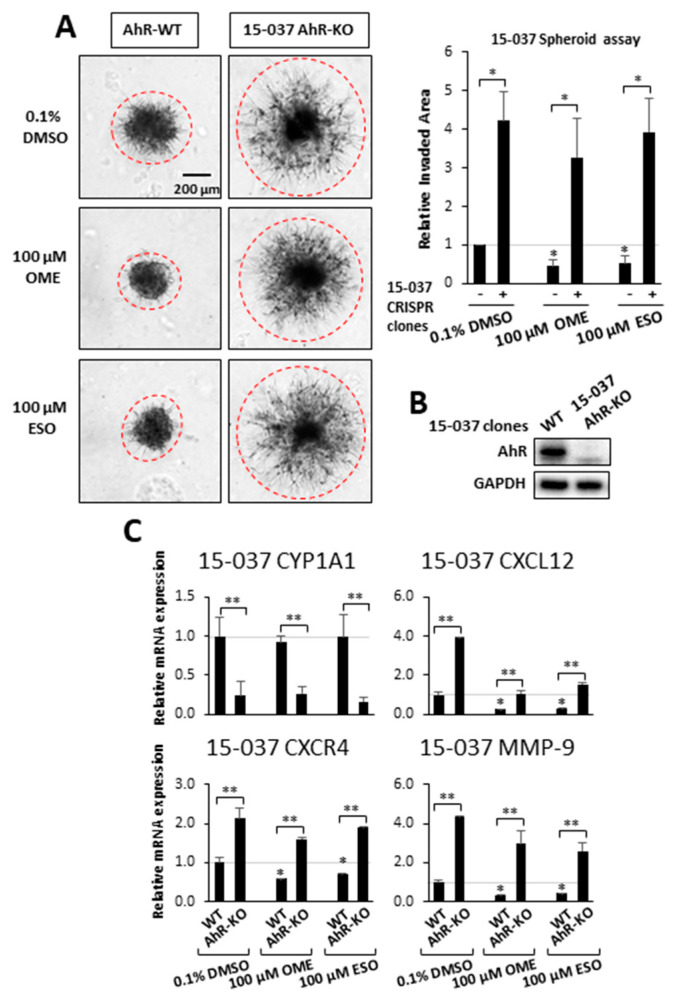
Effects of OME and ESO on AhR-dependent invasion and gene expression in 15-037 and 15-037 AKO-19 cells. Wild-type 15-037 and 15-037 AKO-19 cells were treated with DMSO, 100 µM OME, and 100 µM ESO and spheroid cell invasion was determined and quantitated (**A**) as outlined in the Materials and Methods. AhR expression in 15-037/15-037 AKO-19 cells were determined by Western blots of whole cell lysates (**B**). Effects of DMSO (set at 1.0), 100 µM OME, and 100 µM ESO on gene expression in 15-037 and 15-037 AKO-19 cells (**C**) were determined by real time PCR as outlined in the Materials and Methods. Results (**A**,**C**) are expressed as means ± SD for at least 3 determination for each treatment group and significant (*p* < 0.05) effects by the compounds alone (*) and attenuation of these response after silencing AhR (**) are indicated.

**Figure 4 cancers-12-02097-f004:**
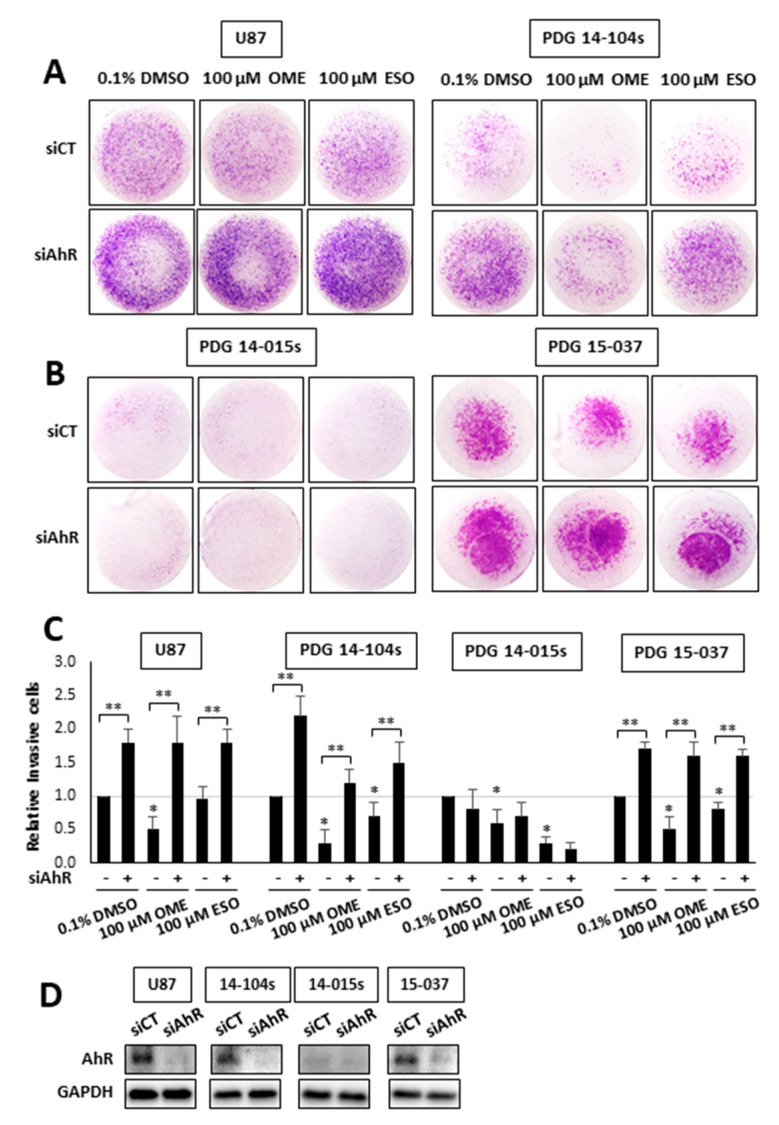
Inhibition of patient-derived GBM cell invasion in the Boyden chamber assay. U87 and 14-014s (**A**), 14-015s and 15-037 (**B**) GBM cells were treated cells were treated with DMSO, 100 µM OME, and 100 µM ESO, and invasion of cells in wild-type (AhR^+/+^) and AhR knockdown cells was determined in a Boyden chamber assay and results were quantitated (**C**) as outlined in the Materials and Methods. Results are expressed as means ± SD for at least 3 separate determinations per treatment group and significant (*p* < 0.05) inhibition by ESO and OME (*) and reversal of these effects after AhR knockdown (**) are indicated. AhR expression after knockdown by siAhR in established and patient-derived GBM cells was determined by Western blot of whole cell lysates (**D**).

**Figure 5 cancers-12-02097-f005:**
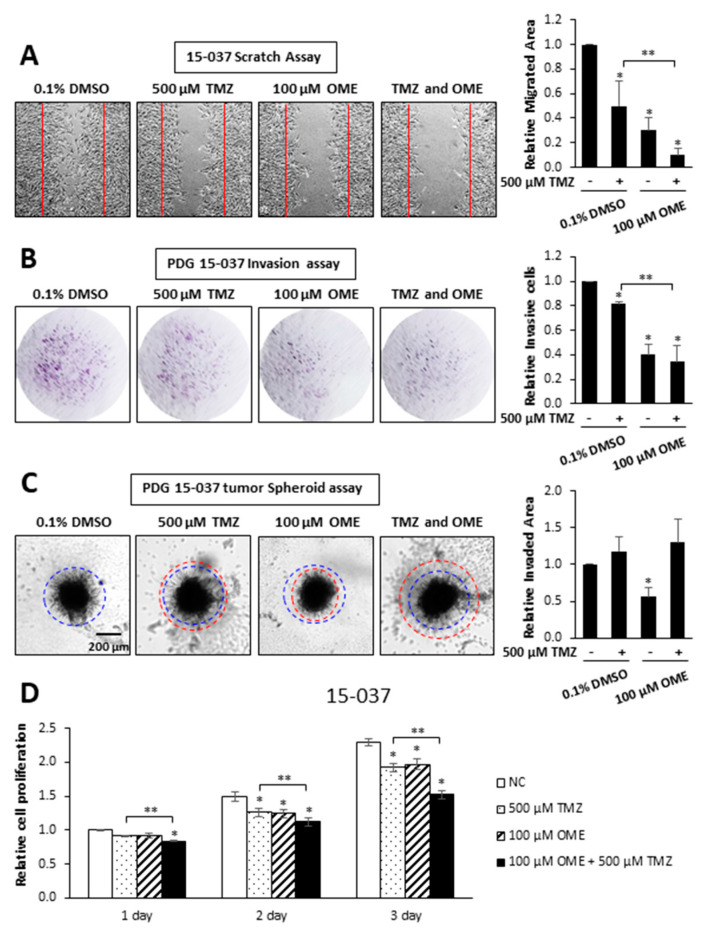
Comparative effects of OME and TMZ on cell migration, invasion and growth. 15-037 cells were treated with DMSO, 100 µM OME, 500 µM TMZ, and their combination and effects on cell migration (**A**), invasion in the Boyden chamber (**B**) and spheroid cell (**C**) assays and proliferation (**D**) were determined as outlined in the Materials and Methods. Results are expressed as means ± SD for at least 3 separate determinations for each treatment group and significant (*p* < 0.05) inhibition by individual compounds (*) and enhanced inhibition by the drug combination (compared to TMZ) (**) are indicated.

**Figure 6 cancers-12-02097-f006:**
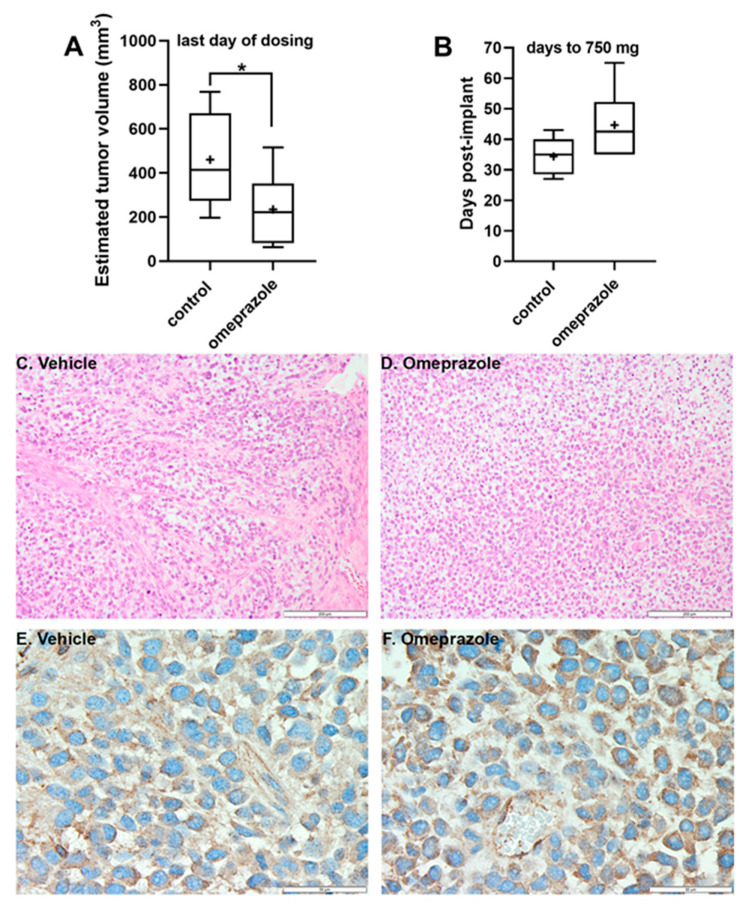
In vivo inhibitory effects of OME. OME inhibited (**A**) and delayed (**B**) tumor growth in the 15-037 GBM PDX tumor model. Tumors from control (*n* = 5) and OME treated (*n* = 6) mice were measured daily. In the box and whisker plots, the horizontal line represents the median, with the mean indicated by the +. The whiskers represent the minimum and maximum. * *p* < 0.05 by one-tailed t-test. H&E staining (**C**,**D**) of representative tumor tissues from control (**C**) and OME mice (**D**) revealed a similar tumor cell density. Scale bars are 200 µM. IHC to visualize AhR (**E**,**F**) demonstrated equivalent cytoplasmic staining in both the control (**E**) and OME mice (**F**). Scale bars are 50 µM.

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
