# Peer review of "Omeprazole Inhibits Glioblastoma Cell Invasion and Tumor Growth"

_cancers, 2020, doi:10.3390/cancers12082097_

Round 1
Reviewer 1 Report
This is an interesting and potentially very important work assessing the impact of selective AHR modulators, in particular the proton pump inhibitor omeprazole, on aggressive growth of human glioblastoma. The paper is well written and structured, the results are accurately presented. However, in larger parts the study is descriptive, lacking a proper mechanistic explanation of the experimental findings, especially with regards to the omeprazole-mediated activation of a specific AHR response. In addition, as outlined below, some of the authors’ observations may not only be due to AHR modulation but also to an interference with other signaling pathways. An obvious point of concern (omeprazole concentration used in the in vitro experiments is approx. 100-fold higher than the peak serum concentration in patients) is adequately mentioned by the authors.
Major points:
- According to the MS title, omeprazole is a ligand of AHR. Has this ever been proven in a competitive AHR ligand-binding assay? I am only aware of a study showing that omeprazole does not compete with TCDD for AHR binding (PMID: 14742684). Please comment and/or correct the MS title accordingly.
- The anti-invasive effects of omeprazole shown in this study are convincing. The authors argue that this is due to a downregulation of invasion-related CXCL12, CXCR4 and MMP-9. Another AHR-activating compound, 3,3’-diindolylmethane, has been previously reported to repress the expression of these genes and associated invasive growth in human breast and ovarian cancer cells (PMID: 18378071). Since CXCL12, CXCR4 and MMP-9 are under transcriptional control of NF-kB-dependent signaling pathways, and 3,3’-diindolylmethane (e.g. PMID: 15665315, 17409440) as well as omeprazole (e.g. PMID: 24048734, 25205289) are known inhibitors of NF-kB, it seems to be possible that the drug-mediated anti-cancer effects observed by the authors are not only due to AHR modulation but also involve, at least to some extent, a disturbance of NF-kB signaling pathways. NF-kB is well-known to drive invasive growth of human glioblastoma and manifold interactions between AHR and NF-kB have been described, including a repression of NF-kB p65 signaling by AHR (e.g. PMID: 25201625, 14678569). Hence, one cannot exclude that an elevated activation of NF-kB in AHR-knockdown cells is responsible for at least some of the observed effects. For example, whereas invasion seems to be marginally affected by omeprazole in the AHR-knockdown cells (Fig. 2A, 4C; see cells 14-104s), omeprazole clearly inhibited the expression of CXCL12 (Fig. 2E, 3C) and MMP-9 (Fig. 2E) not only in AHR-proficient but also in AHR-knockdown cells (the respective contrary statement on page 6, last line, should be corrected). This suggests that also other pathways might be involved in mediating the anti-invasive properties of omeprazole.
The authors should compare NF-kB activity across their AHR-proficient and AHR-deficient cell-lines and treatment groups and check whether the AHR-related pro-invasive effects can be blocked by NF-kB inhibition using a (more potent) specific NF-kB blocker or RNAi.
- According to figure 4, the patient derived cell-line 14-015s appears to be more or less AHR deficient, but does not show an enhanced invasion (fig. 4.B). How does this fit with the other results showing that AHR knockdown increases invasiveness? Please discuss.
Reviewer 2 Report
The authors have shown that omeprazole enhances AhR-dependent inhibition of glioblastoma invasion. The group have already reported similar results of effective treatment of both breast cancer (Jin UH et al. (2014) The aryl hydrocarbon receptor ligand omeprazole inhibits breast cancer cell invasion and metastasis. BMC cancer 14:498. doi:10.1186/1471-2407-14-498) and Pancreatic Cancer (Jin UH et al. (2015) Omeprazole Inhibits Pancreatic Cancer Cell Invasion through a Nongenomic Aryl Hydrocarbon Receptor Pathway. Chemical research in toxicology 28 (5):907-918. doi:10.1021/tx5005198) with omeprazole. As such, the reported results are not novel but can be of significant importance for drug repositioning. As the authors well know, omeprazole inhibits acid secretion and is mainly used for the treatment of peptide ulcers and other gastrointestinal diseases. However, the results of the authors’ group since 2014 suggest that omeprazole can be effective for the treatment of most cancers. It would be more desirable if the authors discuss this point and more generally explain the potential role of omeprazole in cancer treatment.
Reviewer 3 Report
I have read the paper with high interest. It is a paper well written,easy to read and adequate to the journal. I can imagine that the comments
of the others referees are also positive. I recommend the publication.
Author Response
Thank you for your kind comments. All the comments were positive and were taking into consideration with the revised manuscript.